# Soil Fungistasis against *Fusarium Graminearum* under Different tillage Systems

**DOI:** 10.3390/plants12040966

**Published:** 2023-02-20

**Authors:** Skaidrė Supronienė, Gražina Kadžienė, Arman Shamshitov, Agnė Veršulienė, Donatas Šneideris, Algirdas Ivanauskas, Renata Žvirdauskienė

**Affiliations:** 1Lithuanian Research Centre for Agriculture and Forestry, Instituto al. 1, Akademija, LT−58344 Kedainiai Distr., Lithuania; 2Nature Research Centre, Akademijos str. 2, LT−608412 Vilnius, Lithuania

**Keywords:** *Gibberella zeae*, fusarium head blight, suppression, no-tillage

## Abstract

The establishment of the harmful pathogen *Fusarium graminearum* in different agroecosystems may strongly depend on the ability of the soils to suppress its development and survival. This study aimed to evaluate the influence of different soil tillage systems (i.e., conventional tillage, reduced tillage and no-tillage) on soil fungistasis against *F. graminearum*. Soil samples were collected three times during the plant growing season in 2016 and 2017 from a long-term, 20-year soil tillage experiment. The *F. graminearum* in the soil samples was quantified by real-time qPCR. The soil fungistasis was evaluated by the reduction in the radial growth of *F. graminearum* in an in vitro assay. The antagonistic activity of the soil bacteria was tested using the dual culture method. The *F. graminearum* DNA contents in the soils were negatively correlated with soil fungistasis (r = –0.649 *). *F. graminearum* growth on the unfumigated soil was reduced by 70–87% compared to the chloroform fumigated soil. After the plant vegetation renewal, the soil fungistasis intensity was higher in the conventionally tilled fields than in the no-tillage. However, no significant differences were obtained among the tillage treatments at the mid-plant growth stage and after harvesting. 23 out of 104 bacteria isolated from the soil had a moderate effect, and only 1 had a strong inhibitory effect on the growth of *F. graminearum*. This bacterium was assigned 100% similarity to the *Bacillus amyloliquefaciens* Hy7 strain (gene bank no: JN382250) according to the sequence of the 16S ribosome subunit coding gene. The results of our study suggest that the presence of *F. graminearum* in soil is suppressed by soil fungistasis; however, the role of tillage is influenced by other factors, such as soil biological activity, type and quantity of plant residues and environmental conditions.

## 1. Introduction

Over the past decades, the prevailing moldboard ploughing has been widely replaced by extensive tillage systems due to the fact of their economic and environmental advantages. It is well known that extensive tillage has certain benefits over the conventional, such as reduced costs (especially through fuel use), soil erosion, nitrate leaching, increased soil organic matter and activity of soil organisms, improved soil structure and preserved soil moisture [1,2,3,4]. Nevertheless, extensive tillage systems often face problems, including soil compaction [5,6], weed [7,8,9] and disease management [10]. Consequently, noninversion tillage is considered one of the major factors contributing to fusarium head blight (FHB) development, since it causes the increased inoculum resulting due to the presence of a high quantity of host plant residues on the soil surface [11,12,13].

FHB is one of the most harmful cereal diseases, which may result in significant yield losses and grain contamination with mycotoxins [14,15,16]. Several *Fusarium* species may cause this disease, but *F. graminearum* (teleomorph *Gibberella zeae*) has been noted as the main causative agent worldwide, including in Northern Europe in recent decades [17,18]. Certainly, weather conditions are the key factors of FHB development [19]. The pathogen and other insect pest drift to the north and is thought to be promoted by changes in climate and farming practices [13,20,21,22]. The use of extensive tillage, no-tillage and the continuous cropping of cereals, along with climate change, enable the manifestation of FHB on an epidemic scale in many cereal growing regions.

FHB is a typical residue-borne disease [11,13]; therefore, the establishment of *F. graminearum* in different agroecosystems may strongly depend on the soil’s capacity to suppress pathogen survival. The development of suppressive soils is a promising strategy to sustainably and prospectively protect plants against soil-borne diseases [23]. The capability of soil to inhibit germination and growth of fungi is mediated by soil microbiota and is called fungistasis [24]. It is worth noting that the soil fungistasis capacity is closely related to optimal abiotic conditions [25] and the soil microbial community composition and diversity [26,27,28,29,30]. Moreover, the suppressiveness ability of soil to *Fusarium* fungi might be influenced by agricultural practices, such as tillage and cropping systems [9,23,31,32,33]. The effect of different tillage systems on FHB development is widely presented in studies. However, there is a lack of information concerning the influence of tillage on the soil’s suppressive ability against FHB pathogens. Therefore, this study aimed to evaluate the influence of different tillage systems (i.e., conventional tillage, reduced tillage and no-tillage) on the presence of *F. graminearum* in the soil and the determination of soil fungistasis against *F. graminearum*. Additionally, the antagonistic activity of soil bacteria was tested against this fungus, and the most promising strain with the highest suppression toward *F. graminearum* was identified based on the sequence of the 16S ribosome subunit coding gene.

## 2. Results

### 2.1. Quantification of F. Graminearum in Soil

The counting of *F. graminearum* fungal colonies by the dilution plating technique had very poor results, since in the samples of 2016 and 2017, collected both in spring (after the renewal of the plant vegetation) and in autumn (post-harvest), only sporadic colonies of this fungus were detected (Appendix A). The *F. graminearum* DNA quantification by real-time qPCR showed no significant difference in the presence of *F. graminearum* in the soil between years (*p =* 0.1391). However, there were significant differences between the sampling time (*p =* 0.0312) and tillage practices *(p =* 0.0175), with a more than double *F. graminearum* DNA content in spring compared to autumn and no-tillage compared to conventional tillage (Table 1). The amount of *F. graminearum* DNA (pg *F. graminearum* DNA/ng of total DNA) obtained in the soil at each sampling time is presented in Figure 1. The difference between the factor interaction was statistically insignificant. The differentiation of the quantity of *F. graminearum* in the different tillage practices was clearly expressed at each sampling time, except for 2016 after harvesting, when no *F. graminearum* DNA was detected in the soil collected from the RT and NT treatments. The DNA extractions and real-time qPCR reactions for the soil samples from 2016 were performed twice, but the same results were obtained. Accordingly, the seasonal and annual effect was mainly influenced by the reduction in the quantity of *F. graminearum* DNA at this sampling time.

### 2.2. Soil Fungistasis against F. Graminearum

Soil fungistasis was evaluated three times per plant growing season: after the renewal of the plant vegetation at the early stem elongation stage (19 May 2016; 21 April 2017–BBCH 29–33); at the end of the wheat flowering (13 June 2016 BBCH 69) or the development of oilseed rape fruit (19 June 2017 BBCH 77); and after harvesting (27 July 2016 and 29 August 2017). *F. graminearum* growth on the chloroform fumigated soil increased by 70–87% compared to the unfumigated soil (Figure 2). Based on two-way ANOVA and Tukey’s mean separation tests, soil fungistasis against *F. graminearum* was not influenced by the tillage practices or the soil sampling time in 2016; however, in the factor interaction, the lowest soil fungistasis was detected in the no-till treatment after the renewal of plant vegetation (Table 2; Figure 3A). The following year, the soil fungistasis was significantly higher in the conventional tillage system *(p =* 0.0323), and for the soil sampled at the oilseed rape fruit development stage (*p =* 0.0023), the interaction within factors was insignificant (Table 2; Figure 3B). The soil fungistasis in the different tillage treatments negatively correlated (r = −0.649 *) with the *F. graminearum* DNA contents in the soil.

### 2.3. Antagonistic Activity of Soil Bacteria against F. Graminearum

One hundred and four morphologically distinct bacteria were obtained from the soil samples collected at the early wheat stem elongation stage on 19 May 2016. The antagonistic activity against *F. graminearum* was tested on potato dextrose agar plates. Twenty-three isolates showed from a weak (inhibition zone without mycelial growth <1 mm) to strong (inhibition zone without mycelial growth >3 mm) inhibitory effect on the growth of *F. graminearum* (Table 3, Figure 4). The isolate 12–45, with the highest suppression toward *F. graminearum*, was assigned 100% similarity to the *Bacillus amyloliquefaciens* Hy7 strain (GenBank accession number: JN382250), according to the sequence of the 16S ribosome subunit coding gene.

## 3. Discussion

In this study, two different methods were used to quantify the *F. graminearum* in the soil samples. The classical plating technique on selective media yielded very poor results, so we conclude that the density of this fungus in the soil was very low. The quantitative real-time PCR analysis, which is much more sensitive and specific, allowed for the significant detection and differentiation of *F. graminearum* DNA levels among the treatments tested. Based on these results, significant differences were obtained between the two sampling times (spring—after the renewal of plant vegetation; autumn—after harvesting) and soil tillage practices, especially no-tillage and conventional tillage (Table 1). It should be noted that *Fusarium graminearum* persists in the soil through saprotrophic survival on infected crop residues [35]. This fungus is a poor competitor in soil, and the soil community controls its development; however, the presence of the suitable type of plant residues, such as maize, wheat and oilseed rape, acts as a buffer and can ensure the persistence of the fungus in the soil [36]. In addition to type, the amount of plant residue plays a crucial role in the pathogen’s survival. An increase in the rate of the residue decomposition is directly related to a rapid decrease in the abundance of *F. graminearum* [11]. It is also important to note that wheat residues decompose faster and more thoroughly in the soil than on the surface [37]. Thus, we assume that agricultural practices associated with crop residue management, such as tillage, play an essential role in controlling the initial *F. graminearum* inoculum and its subsequent presence in the soil. Our study revealed a decrease in *F. graminearum* DNA after harvest compared to the renewal period of the plant vegetation, as well as in the conventional tillage system compared to no-tillage. Consequently, it can be partially explained by the amount of residue, i.e., the decrease in the undecomposed plant remains. Notably, the seasonal and annual effect was mainly influenced by the decrease in the amount of *F. graminearum* DNA in 2016 after harvesting. This can be explained by the type of preseeding crop residue which, in this case, was a field pea. It is well known that a lower C:N ratio of legume residues is associated with faster decomposition rates than cereal residues [38,39]. In 2017, when wheat plant residue was present in the field, only a slight decrease in the levels of *F. graminearum* DNA in the soil was observed between the sampling times. The presence of *F. graminearum* in soil may also be affected by climatic conditions, as the plant growing season in 2016 was characterized by one-third less precipitation and days with precipitation compared to 2017 (Table 5).

The development of *F. graminearum* in soil might be suppressed by fungistasis caused by the participation of soil microorganisms [26,33]. The microbial community composition is essential for the development of fungistasis through the production of antifungal compounds [28,30,40]. The soil fungistasis effect on *F. graminearum* mycelium growth might be detected using an in vitro assay [41]. Using this method, Lisboa et al. [33] found that the growth of *F. graminearum* was 88.1% greater (mean growth was 5.2 cm greater) for the soil samples fumigated with chloroform compared to unfumigated. Our study also showed that the growth of *F. graminearum* mycelium was significantly increased (on average 80%) in chloroform fumigated soil compared to natural unfumigated (Figure 2). This suggests that inactivated native soil microbiota due to the fact of chloroform fumigation leads to the development of this fungus in the soil. In addition, during the soil fumigation, some of the nutrients can be released from dead microbial cells, which may also enhance fungus colonization. Bonanomi et al. [25] detected that the addition of labile C substances to sterile soil extracts completely relieved fungistasis, restoring fungal growth. Unfortunately, this was not evaluated during our study.

Although the differences among the tillage methods were not always statistically significant, soil fungistasis negatively affected the presence of *F. graminearum* in the soil. Statistically significant negative correlations (r = –0.649 *) between the soil fungistasis in the different tillage treatments and *F. graminearum* DNA contents in the soil were observed. The study showed that soil fungistasis suppresses the presence of *F. graminearum* in the soil; furthermore, multifactorial studies may help to better understand and manage this soil capacity.

Based on two-way ANOVA and Tukey’s mean separation tests, we obtained that the soil fungistasis against *F. graminearum* was significantly affected by the tillage practices or soil sampling time (Table 2). The soil fungistasis was higher in the conventionally tilled fields compared to the no-till, and it showed moderate suppression in the reduced tillage (Table 2, Figure 3). Notably, this effect was revealed only in the spring (after the renewal of the plant vegetation) in 2016 and similarly in 2017 (Figure 3). Any significant differences were obtained among the tillage treatments at the mid- and last soil sampling times. When comparing the timing of the sampling, the least soil fungistasis was obtained in spring and the highest at the oilseed rape fruit development stage in 2017 and similarly at the wheat mid-flowering in 2016. According to Wu et al. [26], the soil fungistasis capacity is closely correlated with the soil bacterial community composition and diversity. Our results demonstrate that the soil fungistasis became stable immediately after harvesting, at approximately 80%, independent of the year and tillage practice. This could be explained by the fact that plant developmental stages might influence the soil microbial community because of the physiological requirements and the composition of plant exudates that vary with their growth. Similarly, a recent study suggested that the rice root microbiota changed substantially throughout the plant developmental stages, stabilized at the beginning of the reproductive stage and had only minor shifts until rice ripening [29]. Our results are completely opposite to those of Lisboa et al. [33], who found that no-tilled soil inhibited *F. graminearum* growth in vitro much more strongly than conventionally tilled. It is difficult to draw a firm conclusion based on only two studies, but the effect of the tillage practices appears to be indirect rather than enhanced by the interaction of other factors. The differences in the biotic and abiotic factors between these two studies may contribute to the differences in the results. The studies mentioned above were performed in Brazil. Correspondingly, the spread of *F. graminearum*, the composition and activity of soil microorganisms, and the environmental conditions and agricultural practice, including type and quantity of pre-crop residues, differed from our experiment. The relationship between fungistasis and microbial diversity has been well documented in previous studies [26,42,43]; in addition, soil fungistasis highly depends on the presence of the optimal soil abiotic conditions (temperature, moisture, pH, redox potential, etc.) [24]. Many studies have been conducted to better understand the association between microbial diversity and soil disease suppressiveness. For instance, some species of *Streptomyces*, *Bacillus*, *Flavobacterium* and *Pseudomonas* are well-known contributors to the inhibition of numerous soil-borne plant diseases [27,28,31,32]. However, a recent study on screening an extensive collection of field soils for suppressiveness to *Fusarium culmorum* revealed that there was no correlation between the specific rhizobacterial taxa and soil suppressiveness [30]. In microcosms experiments, Bonanomi et al. [25] showed that the quality of the organic amendments is a major controlling factor of soil fungistasis. They observed the different intensities of fungistasis reduction among the 42 plant residues tested. They found that the competition of microorganisms for resources is crucial for inducing fungistasis and, simultaneously, limiting the spread of microbial species in soil. The findings supporting their conclusion include (i) the dramatic relief of soil fungistasis when the soil was amended with lignin poor but labile C-rich substrates; (ii) the positive correlation between soil respiration and fungal growth; (iii) the 13C NMR results showed a relationship between soil fungistasis and the biochemical quality of the plant residues and provided a quantitative assessment of the time required for fungistasis restoration after organic materials application [25]. Our study did not examine the influence of different organic matter, soil respiration and biochemical quality, so these are not cases that explain the differences in the intensity of fungistasis. However, a temporary decrease in the intensity of the soil fungistasis in the spring and recovery until the end of the plant vegetation suggests that fungistasis requires a certain amount of time to recover after the introduction of organic matter into the soil. In microcosm experiments, fungistasis was restored after a maximum of 20 days, but it probably takes longer under field conditions. Based on the current study, we cannot specify the exact time required for soil fungistasis recovery. Further evaluation of fungistasis dynamics after soil amendment with different plant residues is needed.

Bacteria from the soil of fields with different tillage treatments were isolated and tested for antagonistic activity against *F. graminearum*. The results are consistent with previous studies [44,45,46], where strains of *Bacillus amyloliquefaciens* exhibited antagonistic activity against *Fusarium* species, including *F. graminearum*. Moreover, numerous bacterial strains of *Bacillus* genera, such as *B. subtilis*, *B. cereus*, *B. pumilis* and *B. licheniformis,* are used as biocontrol agents against different *Fusarium* sp. [26,47,48]. Overall, *Bacillus* species are attractive due to the fact of their ability to produce different substances that provide protection and serve as biological agents [49]. It is worth noting that surfactin, fengycin and iturin are among the compounds derived from *Bacillus* that have received the greatest attention in terms of *F. graminearum* research [44,50]. According to Alvarez et al. [51], plant-associated *Bacillus amyloliquefaciens* strains MEP218 and ARP23 are capable of producing all of the earlier mentioned compounds. This might explain the strong inhibitory effect on the growth of *F. graminearum* by the isolate 12–45 belonging to *B. amyloliquefaciens* in the present study. These findings may explain the role of *B. amyloliquefaciens* in the soil of specific ecosystems as a potential biocontrol agent against *F. graminearum*. Evidence that *B. amyloliquefaciens* isolate 12–45 has potential as a future biological control agent for *F. graminearum* encourages further research focusing on biochemical processes to understand the mechanism of action against this plant pathogen. Other bacterial isolates with moderate fungistatic activity detected in this study may also be ecologically significant if they dominate the microbial community and, thus, merit further study.

## 4. Materials and Methods

### 4.1. Study Site and Experiment Description

The study was carried out at the Institute of Agriculture, Lithuanian Research Centre for Agriculture and Forestry. The long-term tillage experiment was established in 1956 in the central part of Lithuania (55°23’50” N, 23°51’40” E). However, some tillage treatments have changed over the years, and the experimental design has been stable since 2003 (Table 4). The experimental design consisted of four blocks, with three different tillage treatments per block (field replicate): conventional tillage (ploughing), reduced tillage (harrowing) and no-tillage (direct drilling). The gross area of each tillage plot was 22.0 × 10 m. According to the WRB (FAO), system soil is defined as an *Endocalcari-Epihypogleyic Cambisol* of a loam texture. The main soil characteristics in the 0–10 cm surface layer were sand = 47.2%, silt = 35.1%, clay = 17.7%, organic carbon = 1.28%, humus 2.21%, P_2_O_5_ = 256 mg kg^–1^, K_2_O = 272 mg kg^–1^, total N = 0.152%, pH = 7.0 and field capacity = 0.31 m^3^ m^–3^.

The crop rotation consisted of 5 members that have been stable since 2010: field pea (2010, 2015 and 2020), winter wheat (2011, 2016 and 2021), winter oilseed rape (2012, 2017 and 2022), spring wheat (2013 and 2018) and spring barley (2014 and 2019).

### 4.2. Collection of Soil Samples

Soil samples for laboratory analyses were collected from the three tillage treatments (CT, RT and NT) × 4 blocks (total of 12 samples per sampling time) in 2016 and 2017. The soil sampling was performed three times each year during the plant growing season: after the renewal of the vegetation at the early stem elongation stage (19 May 2016 and 21 April 2017–BBCH 29–33), at the end of the wheat flowering (13 June 2016 BBCH 69) or development of oilseed rape fruit (19 June 2017 BBCH 77), and after harvesting (27 July 2016 and 29 August 2017). A tubular soil sampler of 20.0 cm in length and 2.0 cm in diameter was used for the soil collection. The soil samples were taken from the superficial top 10 cm as 6 subsamples per plot, which were then pooled together to obtain one ~300 g sample per field replicate. The soil samples were immediately sieved (3 mm), well mixed and the subsamples of ~50 g were stored at -20 °C until the DNA extraction; the remaining samples (~250 g) were stored at +4 °C for 1–3 days until the soil fungistasis and CFU counting analyses.

### 4.3. Quantification of *F. Graminearum in Soil*

The quantitative assessment of the *F. graminearum* in the soil was performed two times over the plant growing season (renewal of the vegetation and after harvesting) in both years. For the enumeration of the *F. graminearum* in the soil, the dilution plating technique was used. Ten grams of soil were diluted with 90 mL of distilled H_2_O, and 1 mL of this suspension was used to prepare a dilution series (1:10) up to 10^–4^. The three last dilutions were inoculated to empty plates in triplicate, poured with potato dextrose agar and incubated at 25 °C. The total number of fungal colonies and *F. graminearum* colonies were counted after 3, 5 and 7 days of incubation and expressed in colony-forming units per gram of soil (cfu g^–1^). For the detection of the *F. graminearum* CFUs, all colonies morphologically related to *Fusarium* spp. were subcultured, purified, and identified based on the colony morphology and spore shape [52]. The colonies identified as *F. graminearum* were then confirmed by species-specific PCR [53]. 

The DNA for the polymerase chain reactions (PCRs) was extracted from 500 mg of soil using the FastDNA^TM^ SPIN Kit for soil (MP Biomedicals). The procedure was performed according to the manufacturer’s instructions. The soil samples were homogenized using FastPrep 24™ 5G homogenizer (MP Biomedicals, Santa Ana, CA, USA) for 40 s at a speed setting of 6.0. The DNA concentration was measured using Biophotometer (Eppendorf, Germany).

*F. graminearum* was detected in the soil samples by quantitative real-time PCR (qPCR) using primer sets for amplification of the EF1α gene sequences: FgramB379fwd CCATTCCCTGGGCGCT and FgramB411rev CCTATTGACAGGTGGTTAGTGACTGG [54]. qPCR was carried out in a total volume of 12.5 μL: 6.25 μL 2 × SYBR Green PCR Master Mix (Applied Biosystems), 300 nM of each primer (Metabion International AG), 0.4μg/μL bovine serum albumin (BSA) (Thermo Fisher Scientific) and a 2.5 μL ten-fold diluted template DNA. PCR was performed in duplicate for all samples. The reactions were carried out on the 7900 HT Sequence Detection System (Applied Biosystems, Waltham, MA, USA). Cycling protocol: 2 min at 50 and 95 °C for 10 min, followed by 40 cycles at 95 °C for 15 s and 62 °C for 1 min, followed by dissociation curve analysis at 60 to 95 °C. The standard curve was made of ten-fold dilution series using pure *F. graminearum* DNA (7.5–R^2^ = 0.984). The amount of fungal DNA in the soil sample was calculated from the cycle threshold (Ct) values using the standard curve and expressed as pg *F. graminearum* DNA/ng of total DNA. Because SYBR Green binds to all double-stranded DNA, the results of each individual sample were evaluated by studying the dissociation curve and Ct value. The efficiency (E = 10(–1/slope)–1) was calculated from the slope of the linear relationship of the log10 values of the DNA concentration and the cycle number (Ct).

### 4.4. Soil Fungistasis

To determine the effects of the soil fungistasis on *F. graminearum* under different tilling systems, an in vitro assay was performed [33,41]. A completely randomized experimental design included four biological (field) and three technical (laboratory) replicates per tillage treatment. Fifty grams of chloroform-fumigated or unfumigated soil from the samples stored at +4 °C, as described above, were placed in sterilized Petri dishes (Ø 15 cm) (in three technical replicates). The soil surface was smooched, easily compressed and covered with 30 mL of sterile (~50 °C) water–agar. The water–agar acted as an interface between the soil and *F. graminearum*. When the agar stained, the center of the plates was inoculated with potato dextrose agar plugs (8 mm) of actively growing *F. graminearum* mycelium. The plates were incubated in the dark at 22 °C for 4 days, after which the diameter of the fungal colonies was measured. The percentage of fungistasis was calculated as follows:$$\mathrm{Fungistasis}\;(\%)=\frac{\mathrm{unfumigated}\;\mathrm{diameter}-\mathrm{fumigated}\;\mathrm{diameter}}{\mathrm{fumigated}\;\mathrm{diameter}}\times 100$$

### 4.5. Soil Fumigation

The soil samples were weighed into 100 mL plastic beakers and placed in a glass vacuum desiccator for exposure to CHCl_3_. Approximately 50 mL of ethanol-free, hydrocarbon-stabilized CHCl_3_ was poured into a beaker containing boiling chips and placed into a desiccator. A paper towel moistened with deionized water was also placed in the desiccator to maintain the water content of the soils during fumigation. The desiccator was evacuated until the CHCl_3_ boiled. Room air was permitted back into the desiccator, and the process was repeated one more time to promote the distribution of the CHCl_3_ vapor into the micropores of the soil samples. The desiccator was left evacuated for 72 h [55].

### 4.6. Isolation of Soil Bacteria and Evaluation of Antagonistic Activity against F. Graminearum

The dilution method described above was also used to isolate bacteria from the soil. However, the soil suspension was diluted to 10^–6^, and Petri dishes were poured with plate count agar (PCA, Merck) medium. The colonies of the morphologically different bacteria were transferred to PCA plates with after incubation for 12, 24 and 72 h at 28 °C. The antagonistic activity of the bacteria was tested using the dual culture method [26]. The purified soil bacteria were inoculated on PDA medium at the four opposite edges of the Petri dish (2.5 cm from the center of the dish), not in dashes as indicated by the authors but in a dotted manner. A 5 mm^2^ plug of *F. graminearum* mycelium was placed in the center of the plate and incubated at 28 °C. After five days, the diameter of the transparent area around the bacterial culture was measured. The antagonistic activity was assessed based on the average of four replicates assigned to one of three categories: + weak effect—clearly visible inhibition zone without mycelial growth <1 mm in diameter; ++ moderate inhibition—a clearly visible inhibition zone without mycelium growth from 1 to 3 mm in diameter; and +++ strong inhibition—≥3 mm diameter, clearly visible inhibition zone without mycelial growth [34].

### 4.7. Identification of Antagonistic Bacteria

Genomic DNA was extracted from 1 day old bacterial cells using ZR Fungal/Bacterial DNA MiniPrep kit (Zymo Research, Irvine, CA, USA), according to the manufacturer’s instructions and the FastPrep 24™ 5G (MP Biomedicals, Santa Ana, CA, USA) sample homogenizer. The bacterial 16S ribosomal gene sequences for the species identification were amplified using universal primer pairs [56]. All reactions were conducted in mixtures containing 2.5 μL of 10 × PCR buffer (provided with polymerase, Applied Biosystems, Waltham, MA, USA), 2 μL of 2.5 mM dNTP (Thermo Fisher Scientific Baltics, Lithuania), 0.5 μL of each 25 μM primer, 1.25 units of AmpliTaq Gold polymerase (Applied Biosystems, Waltham, MA, USA), 1 μL of extracted DNA template and nuclease-free water up to a total volume of 25 μL. The thermocycling conditions were initial denaturation and polymerase activation at 95 °C for 11 min, then 38 cycles of (95 °C for 40 s, 58 °C for 1 min, 72 °C for 55 s), followed by a final extension at 72 °C for 10 min. Electrophoresis was carried out in a Cleaver MultiSub Mini apparatus (Cleaver Scientific Ltd., UK). The PCR products were observed in 1% agarose gels using the GeneRuler 100 bp DNA Ladder (Thermo Fisher Scientific Baltics, Lithuania) as a marker. The PCR products were sequenced at BaseClear (the Netherlands). The identified nucleotide sequences were processed and analyzed using the DNASTAR program package and the NCBI BLAST database.

### 4.8. Meteorological Conditions

The mean annual temperature and precipitation were obtained from the Lithuanian hydrometeorological station located in Akademija Kedainiai district, Lithuania (the station is located less than 1 km from the study site). The air temperature in April–August 2016 was higher than the monthly long-term (1924–2016) average from 0.3 °C (in August) to 2.7 °C (in May). The amount of precipitation was low in May (only 53% of the long-term average) but heavy in July and August (Table 5). In total, 294.8 mm or 140% of the long-term average precipitation fell during the summer. In most months, the air temperature in the same period of 2017 was close to the long-term average (±0.3–0.5 °C), except July, which was 1.0 °C cooler. May was extremely dry, with only 3.4 mm of rainfall. In the summer, the precipitation was 279.1 mm (long-term average: 212.3 mm).

### 4.9. Statistical Analyses

The statistical analyses were performed using statistical software SAS 9.4 (SAS Institute, Cary, NC, USA). For the analysis of *F. graminearum* DNA and soil fungistasis data, three-way and two-way ANOVA and Tukey’s mean separation tests were performed to test the significance of the differences among the treatments (*p* ≤ 0.05). The correlation coefficient between the mean of the *F. graminearum* DNA and soil fungistasis was calculated using the Pearson correlation test.

## 5. Conclusions

Conventional tillage showed an advantage over reduced tillage and, especially, no-till practices in terms of the presence of *F. graminearum* in the soil, as significantly higher contents of *F. graminearum* DNA (on average 31.5 pg *F. graminearum* DNA/ng of total DNA) were detected in the no-tilled soils compared to the conventional (on average 6.9 pg *F. graminearum* DNA/ng of total DNA). *F. graminearum* growth on the chloroform fumigated soil was significantly increased (on average 80%) compared to the unfumigated soil, which shows the natural capacity of the tested loamy textured *Endocalcari-Epihypogleyic Cambisol* soil to suppress pathogen survival under specific biotic and abiotic conditions. Based on two-way ANOVA and Tukey’s mean separation tests, the soil fungistasis against *F. graminearum* was significantly affected by the tillage practices or soil sampling time. Nevertheless, the overall results of the soil fungistasis suggest that only in spring after the renewal of the plant vegetation was the soil fungistasis intensity higher in the conventionally tilled fields than in the no-till, as no significant differences were obtained among the tillage treatments at the mid- and last soil sampling times. In the different soil tillage treatments, the *F. graminearum* DNA contents were negatively correlated with the soil fungistasis (r = –0.649 *), showing that the relationship between the fungistasis intensity and DNA content decreased. Further evaluation of fungistasis dynamics after soil amendment with different plant residues is needed.

## Figures and Tables

**Figure 1 plants-12-00966-f001:**
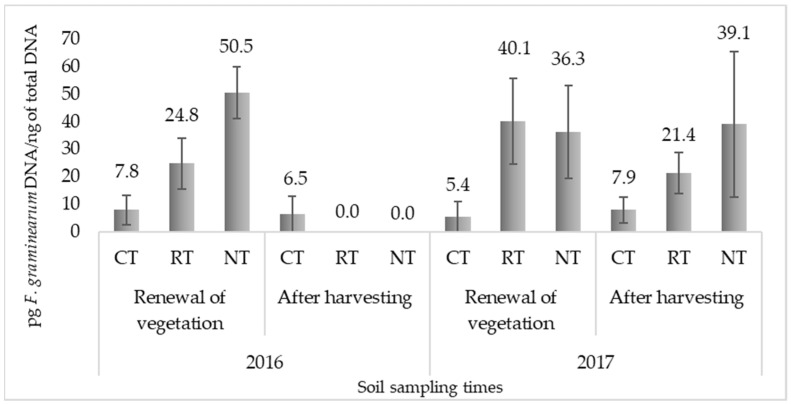
*F. graminearum* DNA (pg *F. graminearum* DNA/ng of total DNA) content obtained in the differently tilled soil at the renewal of plant vegetation (19 May 2016 and 21 April 2017) and after harvesting (27 July 2016 and 29 August 2017) in 2016 and 2017. The error bars indicate the standard deviations within the biological replications at each treatment. CT—conventional tillage (ploughing 22–24 cm); RT—reduced tillage (harrowing: 8–10 cm); NT—no-tillage (direct drilling).

**Figure 2 plants-12-00966-f002:**
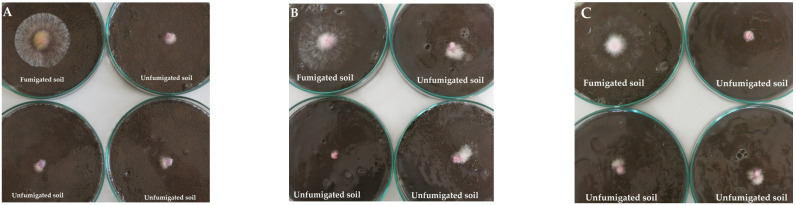
*F. graminearum* growth after 4 days post-inoculation with mycelial plug (Ø 8) mm and incubation in the dark at 22 °C on the fumigated (left upper plate) and unfumigated soil collected from (**A**) conventional tillage, (**B**) reduced tillage and (**C**) no-tillage systems.

**Figure 3 plants-12-00966-f003:**
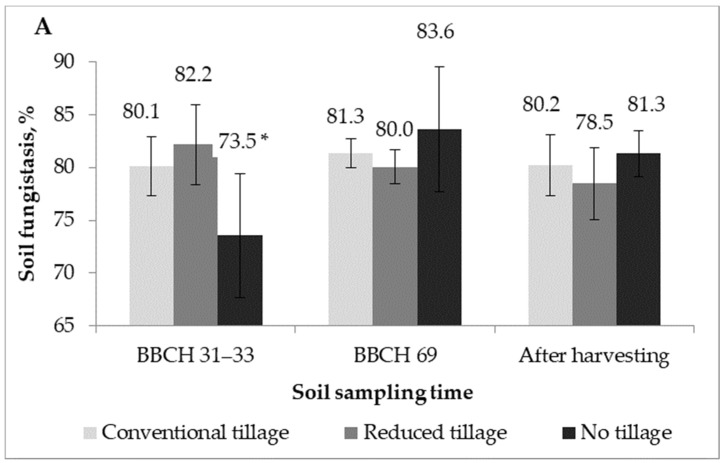
Soil fungistasis (%) against *F. graminearum* growth depending on the different tillage practices and soil sampling times in (**A**) 2016 and (**B**) 2017. BBCH 31–33—renewal of the plant vegetation (19 May 2016 and 21 April 2017); BBCH 69—the end of the winter wheat flowering (13 June 2016); BBCH 77—the development of oilseed rape fruit (19 June 2017) and after harvesting (27 July 2016 and 29 August 2017). * Indicates a significant difference between the treatments at a confidence level of 0.95. The error bars indicate the standard deviations within the biological replications at each treatment.

**Figure 4 plants-12-00966-f004:**
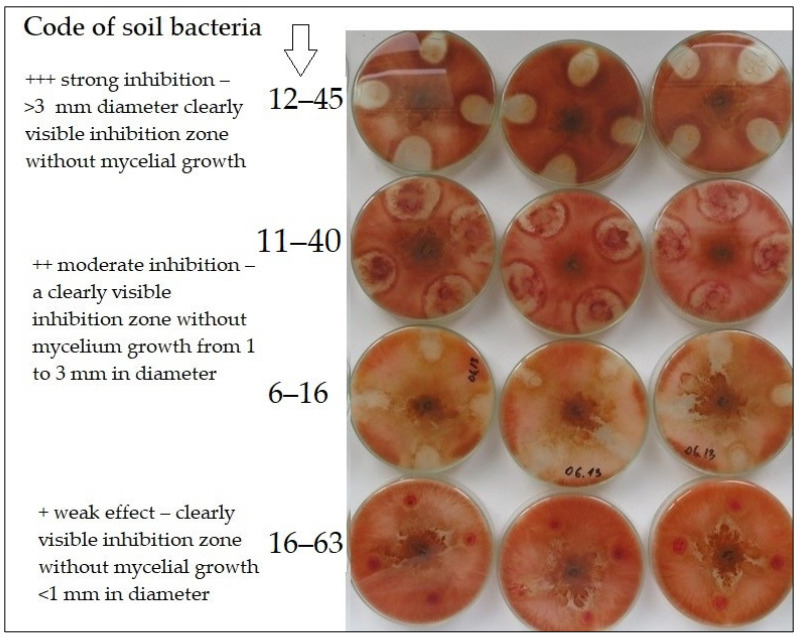
Inhibitory effect of the soil bacteria (isolate codes 12–45, 11–40, 6–16 and 16–63) toward *F. graminearum* in a dual culture on potato dextrose agar plates after 5 days of incubation at 28 °C. A 5 mm agar plug of *F. graminearum* was inoculated on the center of PDA plate, and the bacteria were inoculated on 4 sites of the PDA plate at an equal distance from each other, 2.5 cm apart from the fungus.

**Table 1 plants-12-00966-t001:** Effect of year, sampling time and tillage practice on the mean value of *F. graminearum* DNA (pg *F. graminearum* DNA/ng of total DNA) content in the soil obtained by three-way ANOVA and Tukey’s mean separation test.

	Treatment	Mean	F–Act.	Probability
Year	2016	14.9	2.3	0.1391
	2017	25.0		
Sampling time	Renewal of vegetation	27.5 *	5.07 *	0.0312
	After harvesting	12.5 *		
Tillage practice	Conventional tillage	6.9 **	4.58 *	0.0175
	Reduced tillage	21.6		
	No tillage	31.5 *		

*, ** Indicate the significance of the treatment effect within each tested factor (year, sampling time and tillage practice) at a confidence level of 0.95 and 0.99, respectively. The interactions between the factors were insignificant.

**Table 2 plants-12-00966-t002:** Comparison of the soil fungistasis (%) against *F. graminearum* growth depending on the tillage practice (Factor A) and the soil sampling time (Factor B), obtained by two-way ANOVA and Tukey’s mean separation tests.

	Treatment	Mean, 2016	Mean, 2017
Factor A	Conventional tillage	80.6	82.3 *
	Reduced tillage	80.2	79.7
	No-tillage	79.5	79.4
F–act. Probability		1.89	3.97 *
		0.1726	0.0323
Factor B	BBCH 29–33	78.6	78.0 **
	BBCH 69/77	81.7	82.5 **
	After harvesting	80.0	80.9
F–act. Probability		0.24	7.92 **
		0.7886	0.0023
F–act. A × B Probability A × B		3.38 *	2.27
		0.0250	0.0909

BBCH 31–33—renewal of the plant vegetation (19 May 2016 and 21 April 2017); BBCH 69—the end of the winter wheat flowering (13 June 2016); BBCH 77—the development of oilseed rape fruit (19 June 2017) and after harvesting (27 July 2016 and 29 August 2017). *, ** Indicate the significance of the treatment effect at a confidence level of 0.95 and 0.99, respectively.

**Table 3 plants-12-00966-t003:** Antagonistic activity of soil bacteria against *F. graminearum* mycelial growth on potato dextrose agar plates.

No.	Isolate Code	Soil Sampling Details	Antagonistic Activity
1.	1–1	Conventional tillage	+
2.	2–6	Conventional tillage	+
3.	2–7	Conventional tillage	+
4.	2–8	Conventional tillage	+
5.	14–54	Conventional tillage	++
6.	15–60	Conventional tillage	++
7.	16–62	Conventional tillage	++
8.	16–63	Conventional tillage	+
9.	6–16	Reduced tillage	++
10.	6–17	Reduced tillage	+
11.	7–21	Reduced tillage	++
12.	7–25	Reduced tillage	++
13.	8–28	Reduced tillage	++
14.	17–68	Reduced tillage	++
15.	17–70	Reduced tillage	++
16.	18–73	Reduced tillage	+
17.	19–79	Reduced tillage	++
18.	20–84	Reduced tillage	++
19.	11–40	No-tillage	++
20.	12–43	No-tillage	++
21.	12–45	No-tillage	+++
22.	21–86	No-tillage	++
23.	24–101	No-tillage	++

+ Weak effect—clearly visible inhibition zone without mycelial growth < 1 mm in diameter; ++ moderate inhibition—a clearly visible inhibition zone without mycelium growth from 1 to 3 mm in diameter; +++ strong inhibition—≥3 mm diameter, clearly visible inhibition zone without mycelial growth [34].

**Table 4 plants-12-00966-t004:** Experimental design.

Autumn Tillage Treatments	Year of Establishment	Abbreviation
Conventional tillage (ploughing 22–24 cm)	Since 1956	CT
Reduced tillage (harrowing 8–10 cm)	Since 2003	RT
No-tillage (direct drilling)	Since 2003	NT

Seedbed preparation and drilling were performed across the whole trial using a disc drill.

**Table 5 plants-12-00966-t005:** Meteorological observations for the growing seasons of 2016 and 2017 in Akademija.

Month	Average Air Temperature (°C)		Amount of Rainfall (mm)		No. of Days with Rainfall ≥ 0.1 mm		Mean, 1924–2016	
	2016	2017	2016	2017	2016	2017	Air Temperature (°C)	Amount of Rainfall (mm)
April	7.1	5.6	59.5	48.2	16	18	5.9	37.4
May	15.0	12.8	27.3	3.4	5	4	12.3	51.5
June	17.5	15.4	57.4	72.1	10	17	15.7	62.0
July	18.6	16.7	128.2 (87.4)	153.8	16 (13)	15	17.7	76.7
August	17.1	17.3	109.2	53.2 (53.2)	15	12 (12)	16.8	73.6
April–August	15.1	13.6	381.6 (231.6)	330.7 (330.7)	62 (44)	66 (66)	13.7	301.2

The numbers in the brackets indicate the amount of rainfall and the number of days with rainfall ≥ 0.1 mm until the soil sampling after harvesting.

## Data Availability

Not applicable.

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
