# Peer review of "Soil Fungistasis against Fusarium Graminearum under Different tillage Systems"

_plants, 2023, doi:10.3390/plants12040966_

Round 1
Reviewer 1 Report
This is an interesting and well-design study. I have no additional suggestions on the text, methods, or on the experiment design. However, I would like to ask the authors to correct the term "Fusarium DNA content", and also to use other measurement units. This term is not correct in relation to the RT-PCR. Real-time PCR does not provide data on the amount of Fusarium DNA, but only on the number of EF1α gene copies. Therefore, please, replace "the amount of Fusarium DNA" with "the number of copies of Fusarium graminearum genes" in both the MS text and in the figures. Also, these results should be presented in gene copies per g dry soil rather than pg. Please recalculate your qPCR-based data on the Fusarium graminearum.
Please also add standard deviations to figure 1.
Author Response
Please find the response enclosed.

Reviewer 2 Report
The manuscript reports on an experiment for 2 years in which soils were sampled at 3 times per year in experimental plots for studying the consequences of tillage practices on the survival of an inoculum of the plant pathogen Fusarium graminearum. Finally, it confirms the known effect of tillage, but it tries to demonstrate that fungistasis in soils is the phenomenon causing the lower presence of F. graminearum in soils under tillage management compared to non-tillage management.
There are some limits to the methods used that should at least be discussed by the authors. (i) The data on quantification of F. graminearum in soil should be discussed with more cautions because of the apparent lack of biological replication and lack of DNA extraction replications. The specificity of the primer pair should have been checked by sequencing of the amplicons. (ii) In the fungistasis test based on mycelial growth on fumigated and non-fumigated soils, the very low growth in non-fumigated soil led to difficulties in measuring % of increases due to fumigation and the author did not consider the changes in readily available C and N sources due to fumigation.
Specific comments
Lines 68-71: PDA is not a selective medium for F. graminearum since it allows the growth of many fungi. You should have obtained CFU of other fungi and counted them. It could be interesting to give the data cfu/g and to explain that none looked like F. graminearum.
Line 76: even if there is no statistical overall difference between years, the cultures differed; then it could be interesting to also see the effect of season inside each year. Three times of sampling are presented in Material & Methods, why only two are presented here? Explain in the manuscript.
Figure 1: I guess that the data at the intermediate time of sampling are included in the comparison CT-RT-NT as the total DNA pg is about 70 and about 40 for years or season. Please explain this difference. You could give a table with all the data as supplementary material.
Lines 89-90: I would suggest to present at the contrary. For me fumigation is the treatment, and unfumigated the control, then the conclusion is that the treatment (inactivation of the microorganisms) improved the soil colonisation“.
Figure 2: In the figure, almost no colonisation in non-fumigated soil. did you actually observed any differences in diameters measured in fumigated soils. It was probably complicated to measure and then to calculate % of fungistasis. This could be discussed with the part on differences in soil nutritional status.
In the caption you should add information on the treatments (soils) and the number of incubation days.
Table 1: a footnote should be added for defining BBCH29-33 and 69/77.
Figure 3: * pointing significant differences should be added. Caption: add information about error bars.
Line 120: isolates with ++ fungistatic activity could be ecologically significant if they are dominant in the microbial community. Why you did not identify them? At least add a comment in discussion.
Line 132: PDA is not a selective medium of F.g.
Line 135: Based on and not Bested on.
Lines 148-151: unfortunately, this observation is not present in the manuscript. Could climatic condition explain also? Information is given in Materials &Methods but not exploited here in discussion...
Line 160: Actually, fumigation as a disinfection treatment increased the ability of F. graminearum to colonize the soils. Nutrients could also be released by fumigation (see your comment lines 191-193) and this could be a reason why there is a better colonization. Please comment the balance between fungistasis and stimulation by releasing of N and C resources in your observations.
Lines 162-164: This correlation should have been introduced in Results.
Line 167: not so strong - only in 2017 with a difference of 3 % in table 1 for global effect of tillage practice. You should also discuss the data for RT.
Lines 168-169: It is not surprising because there are more F. graminearum DNA in spring than autumn. You should comment.
Lines 181-182: I understand but you tested fungistasis in vitro. Actually, there is two works with two opposite conclusions about effects of tillage. Does it mean that there is no rule? You should comment.
Lines 182-193: In ref 33, Bonanomi tested against Trichoderma and Mucor with synthetic communities and with plant extracts (not 42 plants). This differ from you own experiment. You should comment. The 3 findings originated from publications from by same authors that should be cited.
Line 197: I do not see why does it “confirm”? How?
Line 198: It was not really proved as there is no kinetics. Many other factors differed between spring and autumn.
Line 248: Did you dry and grind the soils. Is it wet or dry weights?
Line 252: How F. graminearum CFU were selected among the total number of fungal colonies? Please add information.
Line 257: if I understand there was no replication of extraction. Extraction yield is a critical is a critical source of variation and this yield could vary with the soil composition and properties.
Author Response
Please find the response enclosed.

Reviewer 3 Report
Dear corresponding author,
the manuscript titled “Soil fungistasis against Fusarium graminearum under extensive 2 tillage system” provides interesting insights into the evaluation of the influence of extensive soil tillage on soil fungistasis against F. graminearum. The manuscript is well-written and organized and the subject reviewed is worthy of being taken into consideration for publication in Plants. However, in my opinion, some minor revisions are required and I provided a list of revisions to the corresponding author. I sincerely hope that my comments could be useful to improve the article quality as well as the Plants' quality. Thank you to consider me for this revision.
Revision list:
1) ENGLISH: I suggest a general revision of the English language in particular about sentence construction and clarity;
2) Line 19: please indicate also in the abstract the number of years of the long-term experiment;
3) ABSTRACT: I suggest to insert at the end of the abstract sentence to resume the importance of the research;
4) Lines 33-35: Sentence not clear, please reword it;
5) Line 44: Please change to something like: “result in significant yield losses and grain contamination with mycotoxins”;
6) Line 55: Please provide a reference for the definition of fungistasis;
7) Lines 68-70: Consider omitting these results or if you want, report them in the supplementary material. I suggest avoiding the sentence “data not shown”;
8) FIGURE 1: Please insert the standard error bars in columns;
9) FIGURE 2: Please insert in the picture some information to better understand its meaning;
10) FIGURE 4: See the comment above, delete the part with plates lids and insert some information to create e more informative picture;
11) Line 135: Change “basted” with “based”;
12) Lines 217-218: Please insert some perspectives about the study of the potential biocontrol agent B. amyloliquefaciens against F. graminearum;
13) Lines 237-245: Please try to reword section 4.2 and make it clearer;
14) Line 251: How did you identify F. graminearum colonies? Just visual observation? Please specify this;
15) SECTION 4.3: About quantification of F. graminearum in soil: did you express pg F. graminearum DNA/ng of total DNA? If yes, please insert this also in figure 1;
16) SECTION 4.4.: The organization of the soil fungistasis experiment is not clear. Please try to reorganize this section;
17) Line 299: why there is something about spore suspension on bacteria isolation?
Best regards
Author Response
Please find the response enclosed.

Reviewer 4 Report
The manuscript entitled as Soil fungistasis against Fusarium graminearum under extensive tillage system brings new value to our knowledge about soil microorganism and their management, However, the manuscript still immature in its current form. I therefore, recommend the authors to rework on it to improve:
The abstract should stand alone
The introduction must be expand to clearly indicate the importance of the topic with more relevant references to be cited
The authors must clearly describe the EXTENSIVE tillage vrs Conventional tillage as this seems to be mixed up!
Figures are not self explanatory and should be prepared with X and Y axis clearly emphasis
English should be improved
Author Response
Please find the response enclosed.

Reviewer 5 Report
This study, which considers long-term data collection, is of great value and credit should be given to the researchers who conducted it. The study examines the effect of extensive tillage systems on soil's ability to suppress the pathogen Fusarium graminearum. The manuscript is well-written, with a logical flow, and the study goals are clearly presented. The introduction and methods sections provide a thorough description of the experiments and necessary data. The results are also well presented, although some of the figures may require further explanations in the captions.
While the authors have made an effort to critically analyze the findings and compare them with similar studies, the discussion section still requires some additional work. My suggestions have been added in the PDF, and in my opinion, the manuscript would be ready for publication after making some minor changes, particularly in the discussion section.
Specifically, the authors should address limitations of the study in the discussion section, such as the potential impact of climate variables or the weather data, and the need for alternative explanations for the findings, particularly in relation to the potential impact of microbial community on the results.

Author Response
Please find the response enclosed.

Round 2
Reviewer 4 Report
The authors with no doubt significantly improved the manuscript entitled 'Soil fungistasis against Fusarium graminearum under extensive tillage system'. However, they should take care of 2 things: the references are not written in order throughout the manuscript and 2. are the authors sure extensive tillage= zero tillage no tillage?
Author Response
Please find the response enclosed
